# *Salmonella* Typhi and *Salmonella* Paratyphi A elaborate distinct systemic metabolite signatures during enteric fever

Elin Nässtrom[1], Nga Tran Vu Thieu[2], Sabina Dongol[3], Abhilasha Karkey[3], Phat Voong Vinh[2], Tuyen Ha Thanh[2], Anders Johansson[4], Amit Arjyal[2], Guy Thwaites[2,5], Christiane Dolecek[2,5], Buddha Basnyat[3], Stephen Baker[2,5,6]*, Henrik Antti[1]*

[1]Department of Chemistry, Computational Life Science Cluster, Umeå University, Umeå, Sweden; [2]Wellcome Trust Major Overseas Programme, Oxford University Clinical Research Unit, The Hospital for Tropical Diseases, Ho Chi Minh City, Vietnam; [3]Oxford University Clinical Research Unit, Patan Academy of Health Sciences, Kathmandu, Nepal; [4]Department of Clinical Microbiology, Umeå University, Umeå, Sweden; [5]Centre for Tropical Medicine, Oxford University, Oxford, United Kingdom; [6]The London School of Hygiene and Tropical Medicine, London, United Kingdom

**\*For correspondence:** sbaker@oucru.org (SB); henrik.antti@chem.umu.se (HA)

**Competing interests:** The authors declare that no competing interests exist.

**Reviewing editor**: Quarraisha Abdool Karim, University of KwaZulu Natal, South Africa

**Abstract** The host–pathogen interactions induced by *Salmonella* Typhi and *Salmonella* Paratyphi A during enteric fever are poorly understood. This knowledge gap, and the human restricted nature of these bacteria, limit our understanding of the disease and impede the development of new diagnostic approaches. To investigate metabolite signals associated with enteric fever we performed two dimensional gas chromatography with time-of-flight mass spectrometry (GCxGC/TOFMS) on plasma from patients with *S*. Typhi and *S*. Paratyphi A infections and asymptomatic controls, identifying 695 individual metabolite peaks. Applying supervised pattern recognition, we found highly significant and reproducible metabolite profiles separating *S*. Typhi cases, *S*. Paratyphi A cases, and controls, calculating that a combination of six metabolites could accurately define the etiological agent. For the first time we show that reproducible and serovar specific systemic biomarkers can be detected during enteric fever. Our work defines several biologically plausible metabolites that can be used to detect enteric fever, and unlocks the potential of this method in diagnosing other systemic bacterial infections.

## Introduction

Enteric fever is a serious bacterial infection caused by *Salmonella enterica* serovars Typhi (*S*. Typhi) and Paratyphi A (*S*. Paratyphi A) (*Parry et al., 2002*). *S*. Typhi is more prevalent than *S*. Paratyphi A globally, with the best estimates predicting approximately 21 and 5 million new infections with each serovar per year, respectively (*Ochiai et al., 2008*; *Buckle et al., 2012*). Both *S*. Typhi and *S*. Paratyphi A are systemic pathogens that induce clinically indistinguishable syndromes (*Maskey et al., 2006*). However, they exhibit contrary epidemiologies, different geographical distributions, and different propensities to develop resistance to antimicrobials (*Vollaard et al., 2004*; *Karkey et al., 2013*). Additionally, they are genetically and phenotypically distinct, having gone through a lengthy process of convergent evolution to cause an identical disease (*Didelot et al., 2007*; *Holt et al., 2009*).

The agents of enteric fever induce their effect on the human body by invading the gastrointestinal tract and spreading in the bloodstream (*Everest et al., 2001*). It is this systemic phase of the disease that induces the characteristic symptoms of enteric fever (*Glynn et al., 1995*). However, the host's

**eLife digest** Enteric fever is estimated to affect over 37 million people every year. Although treatable with antimicrobial drugs, a slow and/or incorrect diagnosis can result in serious and often life-threatening complications.

Enteric fever is the combined name for typhoid fever and paratyphoid fever. While the symptoms of these diseases are indistinguishable, the strains of *Salmonella* bacteria that cause them are genetically distinct. Moreover, the two organisms that cause the disease exhibit different propensities to develop resistance to antimicrobials. It is important, therefore, to be able to distinguish between typhoid fever and paratyphoid fever so that the correct treatment can be prescribed. However, the diagnostic tools available today struggle to discriminate between *Salmonella* Typhi (which causes typhoid fever) and *Salmonella* Paratyphi A (which causes paratyphoid fever).

Now, Näsström et al. have developed a methodology that can determine if an individual is infected by *Salmonella* Typhi or *Salmonella* Paratyphi A, or neither. Rather than trying to detect the bacteria themselves, the test relies on measuring the levels of various metabolites—molecules produced during metabolism—in the blood. Näsström et al. discovered a set of six metabolites that are affected in different ways by typhoid and paratyphoid fever. The next challenge is to develop this approach so it can be used in endemic settings.

reaction to this systemic spread, outside the adaptive immune response, is not well described. There is a knowledge gap related to the scope and the nature of the host–pathogen interactions that are induced during enteric fever that limit our understanding of the disease and prevent the development of new diagnostic tests (*Baker et al., 2010*). An accurate diagnosis of enteric fever is important in clinical setting where febrile disease with multiple potential etiologies is common. A confirmative diagnostic ensures appropriate antimicrobial therapy to prevents serious complications and death and reduces inappropriate antimicrobial usage (*Parry et al., 2011a*; *Parry et al., 2014*). All currently accepted methods for enteric fever diagnosis lack reproducibility and exhibit inacceptable sensitivity and specificity under operational conditions (*Moore et al., 2014*; *Parry et al., 2011b*). The main roadblock to developing new enteric fever diagnostics is overcoming the lack of reproducible immunological and microbiological signals found in the host during infection.

Metabolomics is a comparatively new in infectious disease research, yet some initial investigations have shown that metabolite signals found in biological samples may have potential as infection 'biomarkers' (*Lv et al., 2011*; *Antti et al., 2013*; *Langley et al., 2013*). As *S.* Typhi and *S.* Paratyphi A induce an phenotype via a relatively modest concentration of organisms in the blood (*Wain et al., 1998*; *Nga et al., 2010*), we hypothesized that the host/pathogen interactions during early enteric fever would provide unique metabolite profiles. Here we show that enteric fever induces distinct and reproducible serovar specific metabolite profiles in the plasma of enteric fever patients.

## Results

### Plasma metabolites in enteric fever

To investigate systemic metabolite profiles associated with enteric fever we selected 75 plasma samples from 50 patients with blood culture confirmed enteric fever (25 with *S.* Typhi and 25 with *S.* Paratyphi A) and 25 age range matched afebrile controls attending the same healthcare facility. Mass spectra were generated by an operator that was blinded to the sample group for each of the 75 plasma samples (n = 105 including duplicates) in a random order using performed two-dimensional gas chromatography with time-of-flight mass spectrometry (GCxGC/TOFMS). This GCxGC/TOFMS data resulted in a series of 3D landscapes of preliminary metabolites (*Figure 1*). Following primary data filtering, 988 unique metabolite peaks were retained.

Comparisons to public databases resulted in 178 GCxGC/TOFMS metabolite peaks that could be assigned a structural identity, and a further 62 peaks that could be assigned to a metabolite class. We additionally highlighted 10 metabolites, via manual inspection, that were found in less than 50 of the 75 samples, which had a diagnostic compatible profile. These 10 metabolites were excluded from the

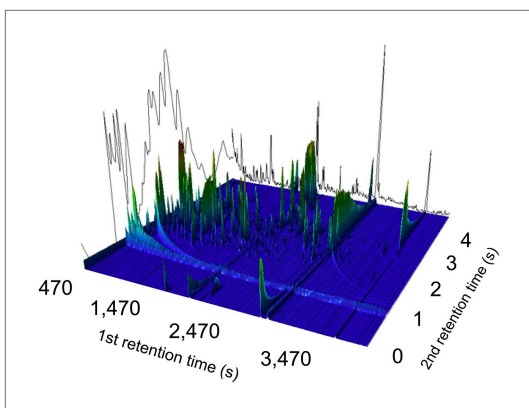

**Figure 1**. A two-dimensional gas chromatogram mass spectrum of a plasma sample from a patient with enteric fever. Image shows a two-dimensional ion chromatogram of unprocessed GCxGC/TOFMS data of a plasma sample from a patient with enteric fever. The three-dimensional landscape depicts detected metabolites peaks in the first dimension (seconds–*x* axis), the second dimension (seconds–*y* axis), and the concentration intensity of the peak signal (*z* axis).

initial pattern recognition modeling, but retained for later analysis. One of these metabolites was found to be significant and was latterly added to the modeling. To further refine the metabolite profiling we aimed to identify profiles that correlated with run order, reducing the risk of instrumental variation into the recognition modeling. We identified 279 metabolites that demonstrated a significant correlation with run order (Pearson coefficient > |0.5|). These 279 metabolites were excluded from initial pattern recognition modeling but still manually investigated. Therefore, 695 unique metabolite peaks (105 samples), were retained for initial pattern recognition modeling.

Principal components analysis (PCA) was used to summarize the systematic variation in the GCxGC/TOFMS data and to generate potential metabolite profiles from the 695 metabolite peaks. We first aimed to identify sample outliers that exhibited extreme metabolite profiles as a consequence of analytical error. We identified 11/105 samples as analytical outliers using PCA. These 11 samples were excluded from further analysis–leaving a total of 94 samples for pattern recognition modeling. These remaining samples were comprised of 32 controls (including analytical replicates of seven samples), 29 *S.* Paratyphi A samples (including analytical replicates of four samples), and 33 *S.* Typhi samples (including analytical replicates of eight samples). Calculation of models excluding all analytical replicates was performed to rule out model overestimation due to replicates; no difference in terms of the model significance was observed.

## Pattern recognition analysis

To investigate the potential of metabolite profiling in enteric fever diagnosis we applied an unsupervised pattern recognition analysis to the filtered metabolite dataset from the cases and controls. The resulting PCA score plot is shown in *Figure 2A*. The variation within the unsupervised pattern recognition model outlined obvious differences between the metabolite profiles in the plasma samples from the controls and the enteric fever patients. It was evident from these analyses that metabolite profiles in the plasma had a potential diagnostic value for enteric fever. However, the samples from patients with *S.* Typhi and *S.* Paratyphi A exhibited substantial overlap, indicating that the metabolite signatures induced by these organisms may be challenging to differentiate.

To obtain a more comprehensive view of the differences between the plasma metabolite profiles between agents of enteric fever we applied a supervised pattern recognition approach. We fitted an extension orthogonal partial least squares with discriminant analysis (OPLS-DA) model to differentiate the GCxGC/TOFMS metabolite profiles in relation to the three sample groups (*Table 1*). The OPLS-DA model generated a $Q^2$ value of 0.45, suggesting reliable differences between the metabolite profiles in relation to the three sample groups. Further validation indicated that the OPLS-DA model provided excellent predictive power for distinguishing between the sample groups (p=1.7 × 10$^{-6}$; control vs *S.* Typhi vs *S.* Paratyphi A). The OPLS-DA method is interpreted through the scores plot (*Figure 2B*); the largest between group differences is found along the first component (t[1]) (x-axis) of the model, while less profound differences are found along the second component (t[2]) (y-axis).

To scrutinize the differences in plasma metabolite profiles between sample groups, new OPLS-DA models were fitted for pairwise comparisons of the sample classes. The score plots for these analyses are shown in *Figure 3* and the summarized data are shown in *Table 1*. As predicted, the OPLS-DA models for differentiating plasma metabolite profiles between samples from the afebrile controls and the two agents of enteric fever exhibited robust and significant separation. The models between the controls and *S.* Typhi infections and between the controls and *S.* Paratyphi A infections also had high predictive power, generating $Q^2$ values of 0.82 (p=4.1 × 10$^{-20}$) and 0.81 (p=4.2 × 10$^{-18}$), respectively

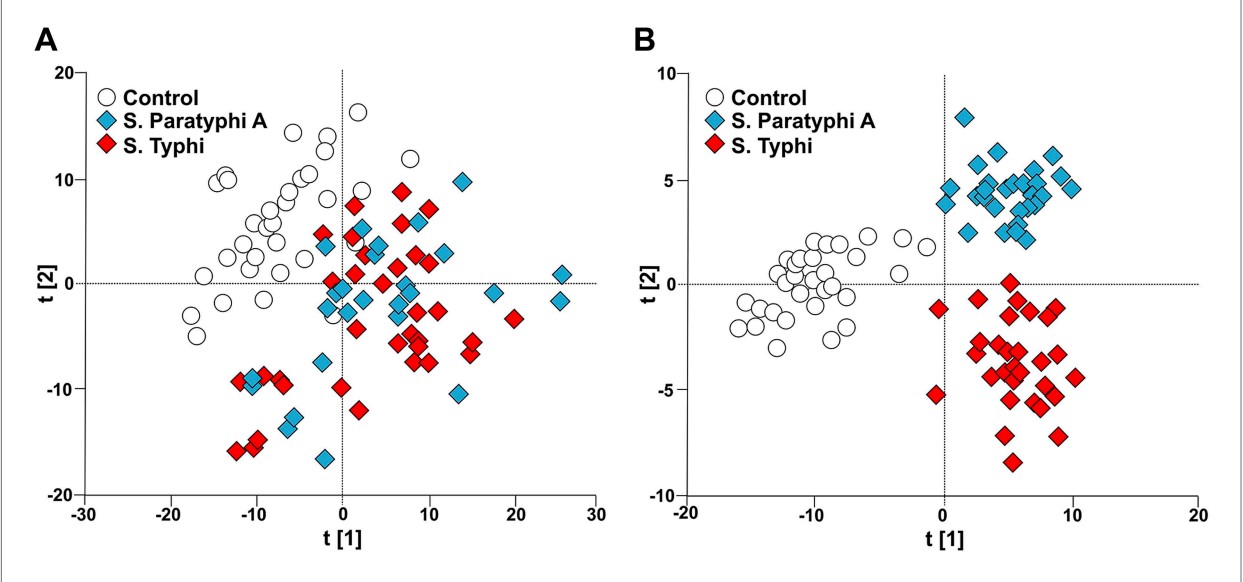

**Figure 2**. Modeling the variation in the GCxGC/TOFMS data in plasma samples from enteric fever patients and controls. (**A**) PCA plot of the first two principal components (t[2] vs t[1]). The PCA plot outlines a separation between the control plasma samples (N = 32; including 7 analytical replicates) and the plasma samples from enteric fever cases (S. Typhi; N = 33 - including 8 analytical replicates, and S. Paratyphi A; N = 29–including 4 analytical replicates). PCA model incorporates 695 metabolites with eight significant principal components ($R^2X = 0.437$, $Q^2 = 0.255$). (**B**) OPLS-DA scores plot of the two predictive components (tp[2] vs tp[1]; x axis and y axis, respectively) outlining a separation between the control plasma samples (N = 32; including 7 analytical replicates) and the plasma samples from enteric fever cases (S. Typhi; N = 33 - including 8 analytical replicates, and S. Paratyphi A; N = 29 - including 4 analytical replicates). OPLS-DA model includes 695 metabolites with two predictive and two orthogonal components ($R^2X = 0.269$, $R^2Y = 0.837$, $Q^2 = 0.451$, p=$1.7 \times 10^{-6}$ [CV-ANOVA]).

(*Figure 3A,B*). The model for differentiating plasma metabolite profiles between the *S.* Typhi infections and the *S.* Paratyphi A infections generated a $Q^2$ value of 0.14 (p=$6.7 \times 10^{-2}$) (*Figure 3C*), indicating that the plasma metabolite profiles can also be used to discriminate between the two enteric fever agents.

Using a combination of the OPLS-DA model variable weights (loadings) and univariate p-values we were able to precisely define the number of metabolite peaks separating the sample groups (*Supplementary file 1*). There were 306, 324, and 58 metabolite peaks separating the controls from the *S.* Typhi infections, the controls from the *S.* Paratyphi A infections, and the *S.* Typhi infections from the *S.* Paratyphi A infections, respectively.

## S. Typhi and Paratyphi A specific metabolites

The presence of 46 metabolites could significantly distinguish between samples from enteric fever cases and control samples, and could also distinguish between samples from *S.* Typhi infected cases and *S.* Paratyphi A infected cases (p≤0.05; two-tailed Student's *t* test) (*Table 2*). Of these 46 informative metabolites, 12 could be annotated. Three metabolites that were found to be significant in all three pairwise OPLS-DA models and annotated (phenylalanine, pipecolic acid, and 2-phenyl-2-hydroxybutanoic acid) were selected for confirmation. The chromatographic profiles of these peaks were compared using the 'raw' GCxGC chromatographic data from one sample in each sample group (*Figure 4*). Phenylalanine and phenyl-2-hydroxybutanoic acid were confirmed to have the highest concentration in the *S.* Typhi sample and the lowest concentration in control sample, while pipecolic acid had the highest concentration in *S.* Paratyphi A samples and the lowest concentration in control samples (*Table 2*). In total, seven metabolites (2,4-dihydroxybutanoic acid, 2-phenyl-2-hydroxypropanoic acid, cysteine, gluconic acid, glucose-6-phosphate/mannose-6-phosphate, pentitol-3-desoxy and phenylalanine) exhibited a higher concentration in the plasma from *S.* Typhi infected patients and five (4-methyl-pentanoic acid, ethanolamine, isoleucine, pipecolic acid, and serine) exhibited a higher concentration in the plasma of *S.* Paratyphi A infected patients (*Table 2*). Of the 34 remaining unidentified

**Table 1.** Multivariate modeling of enteric fever metabolites

| Model * | Number of metabolites included | Number of model components † | R²X ‡ | R²Y ‡ | Q² ‡ | CV-ANOVA § | AUC scores # | AUC CV scores ¶ |
|---|---|---|---|---|---|---|---|---|
| PCA | 695 | 8 | 0.437 | – | 0.255 | – | – | – |
| S. Paratyphi A, S. Typhi, control | 695 | 2 + 2 | 0.269 | 0.837 | 0.451 | $1.7 \times 10^{-6}$ | – | – |
| S. Paratyphi A vs control | 695 | 1 + 2 | 0.261 | 0.961 | 0.815 | $4.2 \times 10^{-18}$ | 1.0 | 0.997 |
| S. Typhi vs control | 695 | 1 + 2 | 0.251 | 0.965 | 0.824 | $4.1 \times 10^{-20}$ | 1.0 | 1.0 |
| S. Paratyphi A vs S. Typhi | 695 | 1 + 1 | 0.160 | 0.714 | 0.140 | $6.7 \times 10^{-2}$ | 0.996 | 0.735 |
| S. Paratyphi A vs control | 46 | 1 + 1 | 0.416 | 0.794 | 0.718 | $8.8 \times 10^{-15}$ | 1.0 | 0.999 |
| S. Typhi vs control | 46 | 1 + 1 | 0.305 | 0.823 | 0.749 | $2.2 \times 10^{-17}$ | 1.0 | 0.996 |
| S. Paratyphi A vs S. Typhi | 46 | 1 + 1 | 0.385 | 0.565 | 0.420 | $2.3 \times 10^{-6}$ | 0.951 | 0.898 |
| S. Paratyphi A vs control | 6 | 1 + 1 | 0.543 | 0.627 | 0.567 | $1.2 \times 10^{-9}$ | 0.964 | 0.948 |
| S. Typhi vs control | 6 | 1 + 0 | 0.299 | 0.529 | 0.492 | $7.6 \times 10^{-10}$ | 0.934 | 0.923 |
| S. Paratyphi A vs S. Typhi | 6 | 1 + 0 | 0.318 | 0.300 | 0.253 | $1.8 \times 10^{-4}$ | 0.801 | 0.796 |

*All OPLS-DA models apart from the highlighted PCA.

†The number of predictive components followed by the number of orthogonal model components.

‡R²X: The amount of variation in X explained by the model, R²Y: The amount of variation in Y explained by the model, Q²: The amount of variation in Y predicted by the model.

§p-value based on cross-validated scores showing the degree of significance for the separation.

#Area under the curve values from receiver operating curves (ROC) calculated from model scores (t).

¶Area under the curve values from receiver operating curves (ROC) calculated from cross-validated models scores (tcv).

metabolites, two were classified as saccharides and exhibited a higher concentration in the plasma of *S.* Typhi patients. We could not assign a structural identity/class to the remaining 32 metabolites (all metabolites summarized in ***Supplementary file 1***).

## Metabolites with diagnostic potential

To investigate the diagnostic potential of the informative metabolites we fitted an OPLS-DA model using the 46 metabolites contributing to the differences between control and infected samples, and between the samples from *S.* Typhi and *S.* Paratyphi A infections (***Table 1***). The model was highly statistically significant for all pairwise comparisons, (p<2.6 × 10⁻⁶; between *S.* Typhi and *S.* Paratyphi A). Furthermore, receiver-operating characteristic (ROC) curves for the fitted and cross-validated OPLS-DA scores for each of the pairwise models verified the diagnostic capabilities of the extracted metabolite profiles (46 metabolites) (area under the curve (AUC) values >0.9 for all comparisons) (***Figure 5***).

The best identifiable metabolite differentiating *S.* Typhi from *S.* Paratyphi A was 2-phenyl-2-hydroxypropanoic acid, which gave an AUC of 0.693 (***Figure 5***), and the best unidentified metabolite differentiating *S.* Typhi from *S.* Paratyphi A gave an AUC value of 0.746. The AUC values for the best individual metabolites differentiating controls from *S.* Typhi infections were 0.884 (phenylalanine) (***Figure 5***) and 0.889 (unidentified), and the AUC values for the individual metabolites best differentiating controls from *S.* Paratyphi A infections were 0.925 (phenylalanine) (***Figure 5***) and 0.926 (unidentified). Finally, we investigated the number of metabolites with confirmed identity or metabolite class required to retain diagnostic power. We found that a metabolite pattern consisting of six identified/classified metabolites (ethanolamine, gluconic acid, monosaccharide, phenylalanine, pipecolic acid and saccharide) gave ROC values >0.8 for all pairwise comparisons (***Figure 6***).

## Discussion

Our work represents the first application of metabolomics to study enteric fever. The potential utility of this method can be observed by the capacity of the metabolite data to successfully identify those with this infection. Currently, the ability to accurately diagnose enteric fever is restricted to a positive microbiological culture result or PCR amplification (***Nga et al., 2010***; ***Parry et al., 2011b***). However, blood culture for suspected enteric fever is commonly only positive in up to 50% of cases only, and

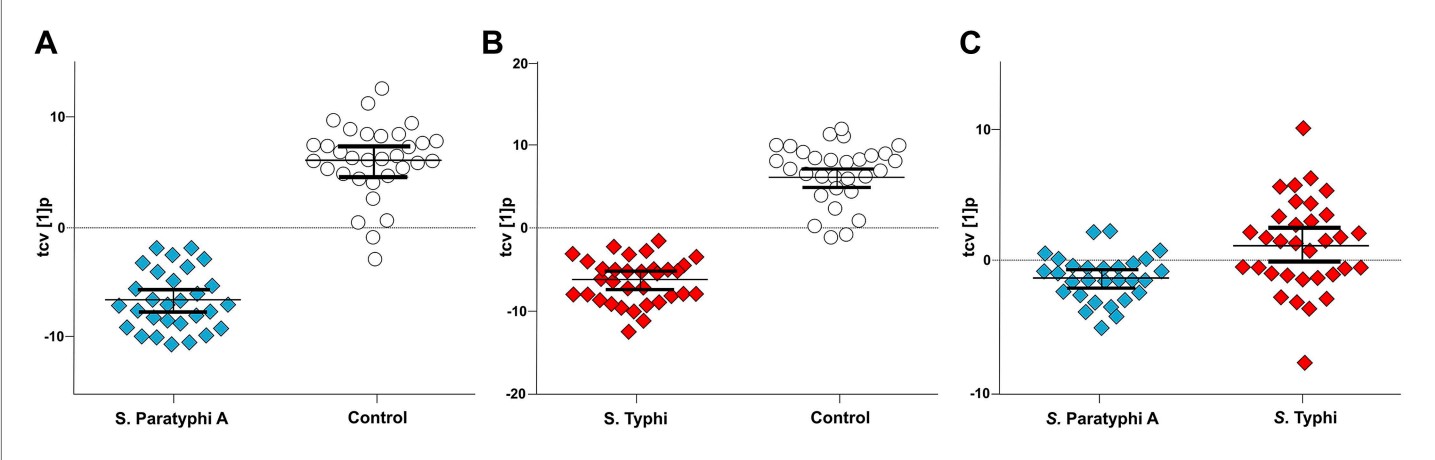

**Figure 3**. Pairwise OPLS-DA models of GCxGC/TOFMS data in plasma samples from controls, *S.* Typhi cases, and *S.* Paratyphi A cases. Cross-validated OPLS-DA scores plots of the first predictive component (tcv[1]p) showing the separation between; (**A**) Controls (N = 32, including 7 analytical replicates) and *S.* Paratyphi A cases (N = 29, including 4 analytical replicates) (p=4.2 × 10$^{-18}$). (**B**) Controls and *S.* Typhi cases (N = 33, including 8 analytical replicates) (p=4.1 × 10$^{-20}$). (**C**) *S.* Typhi cases and *S.* Paratyphi A cases (p=6.7 × 10$^{-2}$). Error bars represent mean score values with 95% confidence intervals. The OPLS-DA model is based on 695 metabolites with one predictive and two orthogonal (**A** and **B**), or one predictive and one orthogonal (**C**) component(s). Additional model information is shown in *Table 1*.

PCR amplification on blood samples performs less well (*Gilman et al., 1975*). In reality, the fundamental complications of enteric fever diagnostics are the low number of organisms in the blood (*Wain et al., 1998*), and a lack of a generic systemic signal. If one combines these limitations with antimicrobial pretreatment and the spectrum of other potential etiological agents circulating in endemic locations, then a substantial technological advance is required to solve the problem of diagnosing enteric fever. It is worth stating that this is a problem worth solving, as enteric fever remains rampant in many low to middle-income countries. Some may argue that the use of broad-spectrum antimicrobials without diagnosis may be prudent. However, this actually compounds the problem, as individuals are often treated with inadequate drugs, inducing treatment failure and facilitating local transmission through fecal shedding (*Parry et al., 2011a*). Furthermore, antimicrobial resistance rates are rising in invasive *Salmonella*, which is associated with treatment failure and complications (*Koirala et al., 2012*; *Walters et al., 2014*).

We found that 306, 324, and 58 metabolites separated the controls from the *S.* Typhi infections, the controls from the *S.* Paratyphi A infections, and the *S.* Typhi infections from the *S.* Paratyphi A infections, respectively. The statistical analyses found that differentiating cases from controls could be performed with considerable power; this was reduced, but still significant, between *S.* Typhi and *S.* Paratyphi A. The majority of distinguishing metabolites among the three groups were unknown, however, some were annotated and had a credible explanation. For example, elevated metabolites distinguishing cases from controls included, 2,4-dihydroxybutanoic acid, phenylalanine, and pipecolic acid. 2,4-dihydroxybutanoic acid is a hydroxyl acid that can be found in low amounts in the blood and urine of healthy individuals, but is also related to hypoxia. Many pathogenic bacteria have the ability to induce the activation of hypoxia inducible factor (HIF)-1 and we surmise that invasive *Salmonella* also play a role in HIF-1 modulation during the inflammatory response induced during early infection (*Werth et al., 2010*). Phenylalanine is an essential amino acid, and higher phenylalanine to tyrosine ratios have been described in the blood of patients with various diseases including sepsis, Hepatitis C (*Herndon et al., 1978*; *Zoller et al., 2012*), and in rats challenged with a number of pathogens (*Wannemacher et al., 1976*). Notably, elevated phenylalanine was also found in during a recent metabolite investigation of primary dengue patients and is intrinsically linked to nitric oxide synthase during infection (*Cui et al., 2013*). Lastly, and most intriguingly, pipecolic acid is a non-protein amino acid and is an essential part of the inducible immunity of plants during challenge from bacterial pathogen and is elevated in the urine of malaria patients (*Sengupta et al., 2011*; *Vogel-Adghough et al., 2013*). These metabolites, which were all elevated in the plasma of enteric fever patients, may be

**Table 2.** Metabolites with discriminatory power for diagnosing enteric fever

| Metabolite * | RT1 † | RT2 † | RI1 † | p-value P vs C | p-value T vs C | p-value P vs T | Change ‡ P vs C | Change ‡ T vs C | Change ‡ P vs T |
|---|---|---|---|---|---|---|---|---|---|
| 2,4-dihydroxybutanoic acid | 1256.4 | 3.22 | 1429.6 | $6.6 \times 10^{-3}$ | $4.9 \times 10^{-4}$ | $4.7 \times 10^{-2}$ | P | T | T |
| 2-phenyl-2-hydroxypropanoic acid | 1724.9 | 2.61 | 1692.6 | $3.7 \times 10^{-2}$ | $1.5 \times 10^{-4}$ | $1.6 \times 10^{-2}$ | P | T | T |
| 4-methyl-pentanoic acid | 627.6 | 2.40 | 1092.8 | $3.1 \times 10^{-2}$ | $5.9 \times 10^{-1}$ | $1.1 \times 10^{-2}$ | P | – | P |
| Cysteine | 1580.0 | 2.96 | 1607.6 | – | $1.7 \times 10^{-2}$ | $3.8 \times 10^{-2}$ | - | T | T |
| Ethanolamine | 880.0 | 3.88 | 1233.6 | $1.2 \times 10^{-3}$ | – | $7.8 \times 10^{-3}$ | P | – | P |
| Gluconic acid | 1985.0 | 0.16 | 1851.7 | $3.3 \times 10^{-2}$ | $1.4 \times 10^{-4}$ | $1.2 \times 10^{-2}$ | P | T | T |
| Glucose-6-phosphate/Mannose-6-phosphate | 2615.3 | 3.65 | 2303.1 | $6.7 \times 10^{-4}$ | $5.9 \times 10^{-5}$ | $4.1 \times 10^{-2}$ | P | T | T |
| Isoleucine | 1012.2 | 3.32 | 1302.9 | $1.1 \times 10^{-2}$ | – | $4.3 \times 10^{-2}$ | P | – | P |
| Monosaccharide_137 | 1622.5 | 4.87 | 1633.8 | $6.0 \times 10^{-3}$ | – | $6.1 \times 10^{-3}$ | C | – | T |
| Pentitol-3-desoxy | 1490.0 | 4.22 | 1557.9 | $4.4 \times 10^{-9}$ | $5.5 \times 10^{-13}$ | $1.1 \times 10^{-2}$ | P | T | T |
| Phenylalanine | 1784.1 | 2.68 | 1728.4 | $3.0 \times 10^{-7}$ | $1.3 \times 10^{-10}$ | $2.4 \times 10^{-2}$ | P | T | T |
| Pipecolic acid | 1130.0 | 3.10 | 1363.1 | $2.4 \times 10^{-5}$ | $2.5 \times 10^{-3}$ | $3.0 \times 10^{-2}$ | P | T | P |
| Saccharide_181 | 2529.1 | 3.99 | 2237 | $1.6 \times 0^{-5}$ | $4.3 \times 10^{-2}$ | $2.7 \times 10^{-2}$ | C | C | T |
| Serine | 1070.0 | 2.60 | 1332.1 | $1.7 \times 10^{-2}$ | – | $4.8 \times 10^{-2}$ | P | – | P |
| Unknown_230 | 549.2 | 2.32 | 1036.8 | $1.7 \times 10^{-3}$ | – | $9.5 \times 10^{-3}$ | P | – | P |
| Unknown_231 | 1090.0 | 2.42 | 1342.3 | $2.8 \times 10^{-3}$ | – | $4.3 \times 10^{-2}$ | P | – | P |
| Unknown_242 | 1550.0 | 2.94 | 1590.5 | $4.0 \times 10^{-5}$ | $1.9 \times 10^{-2}$ | $4.4 \times 10^{-2}$ | C | C | T |
| Unknown_268 | 1895.0 | 3.64 | 1796.1 | – | $1.1 \times 10^{-2}$ | $2.2 \times 10^{-2}$ | – | T | T |
| Unknown_270 | 626.4 | 3.90 | 1093.1 | $1.7 \times 10^{-2}$ | – | $3.2 \times 10^{-3}$ | P | – | P |
| Unknown_281 | 680.0 | 3.38 | 1124.1 | $2.7 \times 10^{-3}$ | – | $3.1 \times 10^{-2}$ | P | – | P |
| Unknown_294 | 725.1 | 2.18 | 1148.5 | $2.1 \times 10^{-3}$ | – | $1.7 \times 10^{-2}$ | P | – | P |
| Unknown_303 | 1900.0 | 2.57 | 1798.5 | $9.1 \times 10^{-3}$ | $1.5 \times 10^{-4}$ | $2.0 \times 10^{-2}$ | P | T | T |
| Unknown_334 | 2790.0 | 2.15 | 2443.5 | $1.9 \times 10^{-5}$ | $2.8 \times 10^{-8}$ | $7.8 \times 10^{-3}$ | P | T | T |
| Unknown_341 | 523.5 | 2.21 | 1018.4 | $2.5 \times 10^{-3}$ | – | $2.7 \times 10^{-2}$ | P | – | P |
| Unknown_364 | 775.1 | 2.25 | 1176.3 | $6.8 \times 10^{-3}$ | – | $2.3 \times 10^{-2}$ | P | – | P |
| Unknown_377 | 961.1 | 2.43 | 1275.6 | $1.9 \times 10^{-3}$ | – | $3.1 \times 10^{-4}$ | P | – | P |
| Unknown_384 | 1010.1 | 2.48 | 1301.4 | $4.9 \times 10^{-3}$ | – | $2. \times 10^{-2}$ | P | – | P |
| Unknown_397 | 1144.9 | 2.75 | 1370.6 | $2.1 \times 10^{-2}$ | – | $2.8 \times 10^{-2}$ | P | – | P |
| Unknown_467 | 1550.4 | 2.92 | 1590.7 | $1.6 \times 10^{-4}$ | $2.4 \times 10^{-2}$ | $4.7 \times 10^{-2}$ | C | C | T |
| Unknown_470 | 1570.0 | 4.02 | 1602.4 | $2.3 \times 10^{-2}$ | – | $1.9 \times 10^{-2}$ | C | – | T |
| Unknown_490 | 1660.6 | 2.27 | 1654.6 | – | $1.1 \times 10^{-3}$ | $2.1 \times 10^{-2}$ | – | T | T |
| Unknown_495 | 1695.0 | 3.26 | 1675.4 | $2.9 \times 10^{-5}$ | $2.3 \times 10^{-6}$ | $2.0 \times 10^{-2}$ | P | T | T |
| Unknown_547 | 1995.0 | 2.33 | 1859.6 | – | $1.2 \times 10^{-2}$ | $3.1 \times 10^{-2}$ | – | T | T |
| Unknown_604 | 2349.5 | 3.27 | 2102.0 | $1.9 \times 10^{-2}$ | – | $3.6 \times 10^{-2}$ | P | – | P |
| Unknown_637 | 2560.7 | 3.99 | 2261.3 | $8.9 \times 10^{-6}$ | $4.5 \times 10^{-3}$ | $4.0 \times 10^{-2}$ | C | C | T |
| Unknown_638 | 2561.3 | 2.67 | 2260.7 | – | $3.2 \times 10^{-3}$ | $7.7 \times 10^{-3}$ | – | T | T |
| Unknown_676 | 2870.0 | 3.28 | 2511.6 | $1.9 \times 10^{-7}$ | $2.5 \times 10^{-3}$ | $4.0 \times 10^{-2}$ | C | C | T |
| Unknown_681 | 2938.1 | 2.75 | 2570.3 | $6.6 \times 10^{-4}$ | – | $5.3 \times 10^{-3}$ | C | – | T |
| Unknown_745 | 770.0 | 3.17 | 1174.0 | $1.6 \times 10^{-3}$ | – | $3.1 \times 10^{-2}$ | P | – | P |
| Unknown_798 | 855.0 | 2.36 | 1219.7 | $4.6 \times 10^{-3}$ | – | $1.0 \times 10^{-2}$ | P | – | P |
| Unknown_811 | 1445.0 | 2.93 | 1532.2 | $3.0 \times 10^{-2}$ | $3.6 \times 10^{-4}$ | $2.5 \times 10^{-2}$ | P | T | T |
| Unknown_914 | 3194.9 | 2.61 | 2802.5 | – | $1.1 \times 10^{-3}$ | $3.2 \times 10^{-2}$ | – | C | P |
| Unknown_949 | 2661.8 | 2.07 | 2339.1 | $7.1 \times 10^{-6}$ | $1.9 \times 10^{-9}$ | $1.3 \times 10^{-2}$ | P | T | T |

*Table 2. Continued on next page*

*Table 2. Continued*

| Metabolite * | RT1 † | RT2 † | RI1 † | p-value P vs C | p-value T vs C | p-value P vs T | Change ‡ P vs C | Change ‡ T vs C | Change ‡ P vs T |
|---|---|---|---|---|---|---|---|---|---|
| Unknown_961 | 2065.4 | 2.71 | 1905.4 | $2.8 \times 10^{-2}$ | $6.6 \times 10^{-4}$ | $2.2 \times 10^{-2}$ | P | T | T |
| Unknown_963 | 1045.1 | 2.32 | 1319.1 | $1.3 \times 10^{-5}$ | $3.6 \times 10^{-5}$ | $4.0 \times 10^{-2}$ | P | T | P |
| Unknown_981 | 2748.2 | 2.05 | 2408.5 | $1.8 \times 10^{-4}$ | $4.9 \times 10^{-8}$ | $2.1 \times 10^{-2}$ | P | T | T |

*Metabolites with statistically significant differences in two or three pairwise comparisons according to univariate p-values (≤0.05) and covariance loadings w* (<|0.03|). T vs C; S. Typhi vs control, P vs T; S. Paratyphi A vs S. Typhi and P vs C; S. Paratyphi A vs controls.

†RT1; 1st dimension retention time (s), RT2; 2nd dimension retention time (s), RI1; 1st dimension retention index.

‡Change in metabolite concentration for each of the pairwise comparisons where P indicates higher concentration in S. Paratyphi A samples, T indicates a higher concentration in S. Typhi samples, and C indicates a higher concentration in control samples.

generic markers of systemic disease and may prove to be vital in determining other bacterial bloodstream infections.

Our data also allowed us to determine different metabolite profiles between those with enteric fever caused by *S.* Typhi and *S.* Paratyphi A. These organisms have a modified physiology in comparison to other *Salmonella* and enter human tissue with limited intestinal replication and by potentially suppressing gastrointestinal inflammation (*Jones and Falkow, 1996*). Consequently, one of the key features of enteric fever is a lack of gastrointestinal involvement as seen with other, non-invasive, *Salmonella* serovars. The majority of the metabolites distinguishing *S.* Typhi from *S.* Paratyphi A may be explained by these subtle biological differences between these organisms and partly by the presence of the virulence (Vi) capsule on the surface of *S.* Typhi, which is absent from *S.* Paratyphi A. Vi is a polysaccharide that has anti-inflammatory properties, limiting complement deposition and restricting immune activation (*Jansen et al., 2011*). The presence and functionality of Vi can be observed in the metabolites differentiating *S.* Typhi from *S.* Paratyphi A as the concentrations of monosaccharide and saccharide were significantly higher in the plasma samples from *S.* Typhi patients than from the *S.* Paratyphi A infections. Conversely, ethanolamine was in significantly higher concentrations in the plasma from the *S.* Paratyphi A patients than in *S.* Typhi patients' plasma. Ethanolamine is released by host tissue during inflammation and experimental work in mice has shown that *Salmonella S.* Typhimurium has a growth advantage in an inflamed gut (*Thiennimitr et al., 2011*). Therefore, the differential detection of ethanolamine in plasma samples from enteric fever patients with different infecting serovars, may be explained by Vi negative *S.* Paratyphi A not having the capacity to control gastrointestinal inflammation to the same extent as *S.* Typhi.

The main limitation of our work was that the samples were restricted to one set of enteric fever cases only. The reason we restricted analysis to enteric fever, rather than a range of bloodstream infections, we because we felt that this was the most robust test for the methodology. Furthermore, as the samples in the study we collected as part of an enteric fever clinical trial we had a range of clinical data and observations on which to link the metabolite profile with. We suggest that future studies in this area are designed to address this limitation, both for validation in different enteric fever cohort and for comparison to other bloodstream infections. The methodology present here should be applied to future 'fever studies' on which there may be a wide array of pathogens. The results from this study leads us to hypothesize that this method could be applied to study the differential metabolite signals between enteric fever and multiple invasive infections and could potentially differentiate between an extensive spectrum of causes of systemic disease or both bacterial, viral, and parasitic etiology. Our work strongly supports this notion, as the metabolite profiles were able to distinguish between those infected with *S.* Typhi and *S.* Paratyphi A, which until now, with the exception of microbial culture has never been a feasible goal. *S.* Typhi and *S.* Paratyphi A have subtle biochemical differences but cause an identical disease syndrome and therefore theoretically induce similar host–pathogen interactions via the adaptive immune response. Consequently, we argue, that whilst our study was limited to enteric fever, the methodology should have the power to distinguish between *Salmonella* and other common bacterial causes of bloodstream infections with more disparate epidemiology, biochemical structure, and pathogenicity (*Nga et al., 2012*).

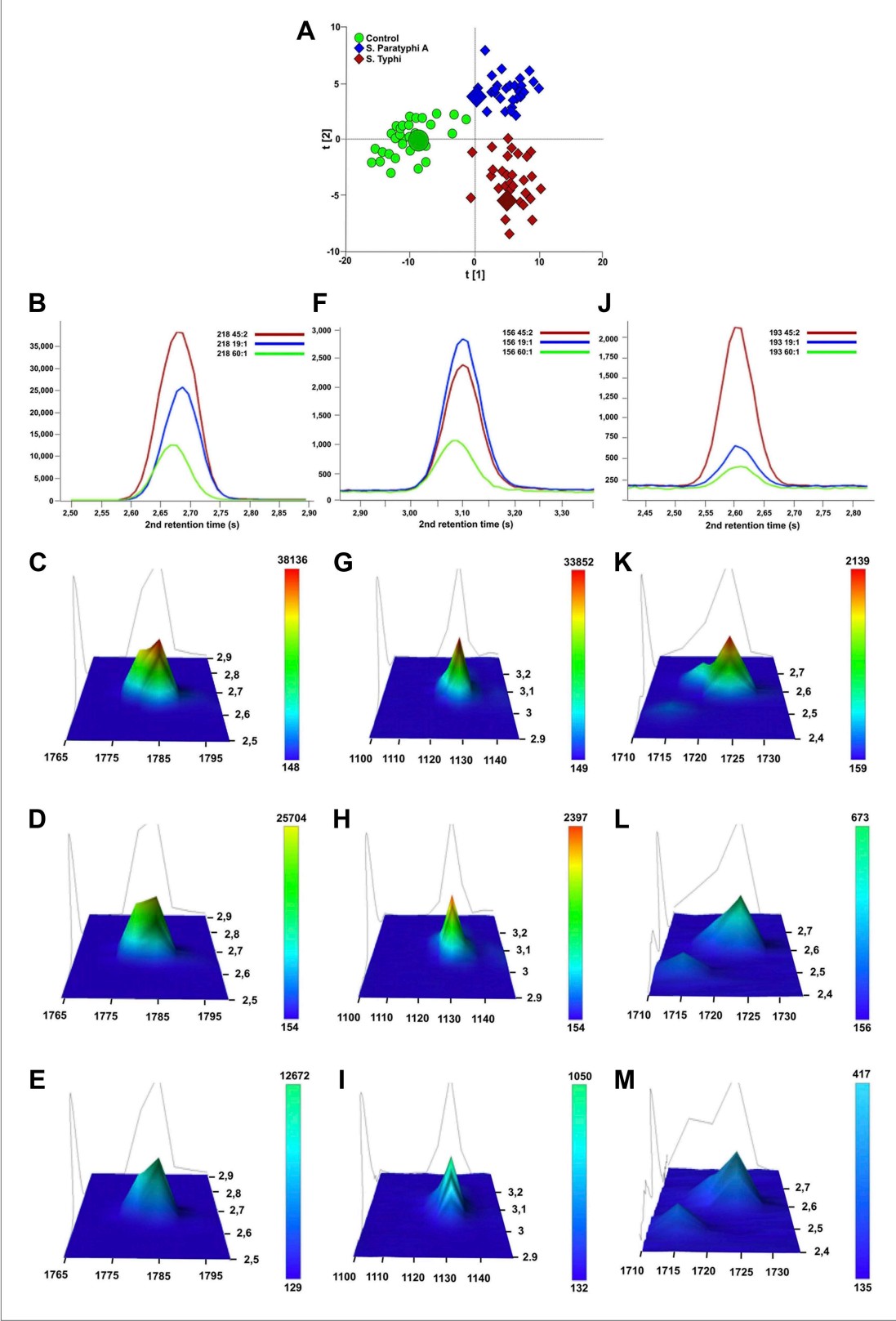

**Figure 4**. Verification of metabolite signals in plasma samples from a control and patients with *S*. Typhi and *S*. Paratyphi A infections. Three metabolites, in three samples from each sample group that were statistically significant in differentiating between sample classes using pattern recognition model-ling, were selected for confirmation using unprocessed chromatographic data. (**A**) OPLS-DA scores plot (tp[2] vs tp[1]) highlighting the three selected
*Figure 4. Continued on next page*

*Figure 4. Continued*

samples (*S.* Typhi: 45, *S.* Paratyphi A: 19, and control: 60). Panel **B**–**D** show one dimensional chromatographic peaks representing each metabolite from the three unprocessed plasma samples (coloured by sample group). Second dimension retention times (s) are shown along the x-axes and the peak intensities along the y-axes. (**B**) Phenylalanine (mass: 218, 1st retention time: 1785 s). (**C**) Pipecolic acid (mass: 156, 1st retention time: 1130 s). (**D**) 2-phenyl-2-hydroxybutanioc acid (mass: 193, 1st retention time: 1725 s). Panel **E**–**M** show the corresponding two dimensional chromatographic peaks with one peak for each sample and metabolite. First and second dimension retention times (s) are shown along the x and y-axes, respectively, and the peak area is shown along the z-axes. The peaks are coloured according to area (colour scale is shown to the right) and the top colour for the two lowest peaks for each metabolite is determined according to the colour scale of the highest peak for the same metabolite. (**E**, **H**, **K**) Phenylalanine for sample 45, 19, and 60, respectively. (**F**, **I**, **L**) Pipecolic acid for sample 19, 4, 5 and 60, respectively. (**G**, **J**, **M**) 2-phenyl-2-hydroxybutanioc acid for sample 45, 19, and 60, respectively.

The science of metabolomics is relatively new, yet this method has previously shown some utility in human disease. In fact, similar methodology has shown potential in generating diagnostic markers for cancer, Dengue fever, Malaria, and *Mycobacterium tuberculosis* (*du Preez and Loots, 2013*; *Sengupta et al., 2011*; *Cui et al., 2013*). This study is the first where the technique has been applied specifically to enteric fever and also, to the best of our knowledge, the first to use two-dimensional gas chromatography/mass spectrometry GC/MS to interrogate plasma for potential biomarkers of infection in human blood. GCxGC/TOFMS offers an exquisite degree of resolution and sensitivity for metabolomics profiling (*Baumgarner and Cooper, 2012*; *Hartonen et al., 2013*). This technique has a substantial methodological advantage over standard GC/MS as it has the ability to span a more expansive proportion of the metabolome, but the resulting data remains compatible with existing mass spectral libraries for metabolite identification. By combining this high-level sensitivity and metabolite identification rate with a multivariate pattern recognition approach we have generated a robust tool for extracting metabolite patterns comprised of structurally identifiable metabolites with diagnostic potential. The extracted metabolite patterns exploit a correlation between relevant metabolites to define a signature that have a greater degree of diagnostic power than any individual metabolite in isolation. The fact that some of the metabolites in the patterns were structurally identifiable, and relatively few, is advantageous in that their biological relevance can be examined and validated as well and their conversion into a practical diagnostic test may be straightforward both in verification and clinical application.

We suggest that the method outlined here could be applied to other diseases with an indistinguishable syndrome of questionable etiology and the validation of these findings and the identification of metabolite signatures induced by other bacterial infections would provide greater confidence and utility. The potential drawbacks of this methodology are cost and portability; we do not advocate that every laboratory in an endemic enteric fever location should invest in a system to support this method. However, a combination of these markers may be suitable for miniaturization into a point-of-care test to measure blood concentrations in suspected enteric fever patients. The format of this diagnostic testing system is currently unclear, but simple lateral flow assays are currently able to detect small concentrations of antigens and other chemicals in whole blood. This approach requires substantial validation and development, yet we predict that the procedure has enough sensitivity to be used on small blood volumes. As an intermediate step we aim to develop this method using small blood volumes and dried blood spots on a range of febrile disease to increase utility in research investigations. A future commercial possibility would be the development of a portable system that associates metabolites in biological samples to a database of metabolites detected during known infections. Indeed, this may not be far away as similar systems are in use for bacterial identification in diagnostic microbiology laboratories (*Marko et al., 2012*).

In summary, we show that reproducible and serovar specific metabolite biomarkers can be detected in plasma during enteric fever. Our work outlines several novel and biologically plausible metabolites that can be used to diagnose enteric fever, and unlocks the potential of this method in understanding and diagnosing other systemic infections.

## Materials and methods

### Ethical approval

The institutional ethical review boards of Patan Hospital and The Nepal Health Research Council and the Oxford Tropical Research Ethics Committee in the United Kingdom approved this study. All adult

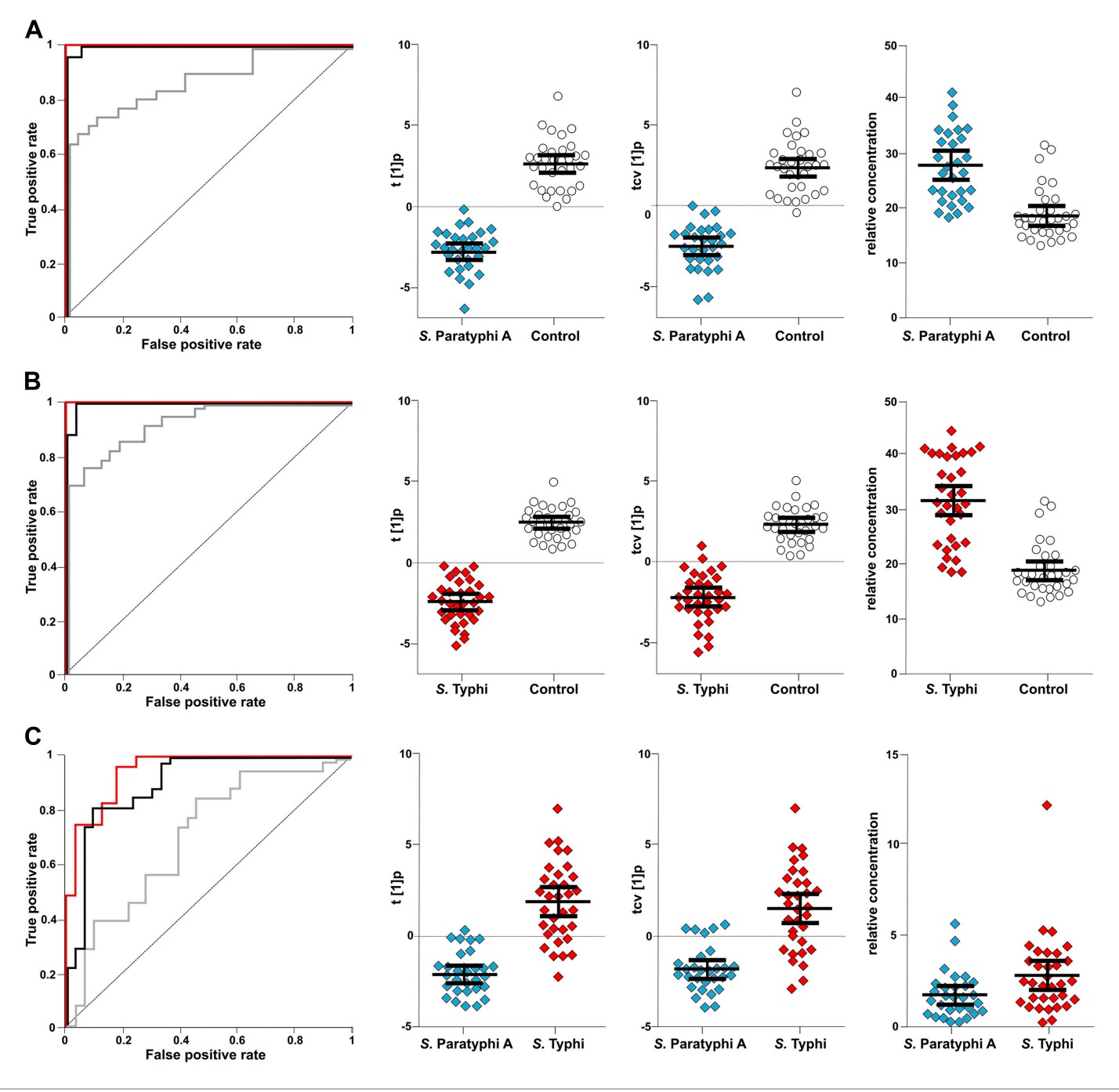

**Figure 5**. The discriminatory power of 46 metabolites to distinguish between plasma samples from controls, *S.* Typhi cases, and *S.* Paratyphi A cases. Panels on the left show the ROC-curves based on scores (red lines) and cross-validated scores (black lines) from OPLS-DA models using the 46 most statistically significant (*S.* Typhi against controls and/or *S.* Paratyphi A against controls) metabolites separating enteric fever samples from control samples and separating *S.* Typhi samples from *S.* Paratyphi A samples. The ROC curve showing the best individual discriminating metabolite is shown by the grey line. The scatterplots show pairwise class differences based on scores (t[1]p) (left), cross-validated scores (tcv[1]p) (centre) from OPLS-DA models using the 46 most statistically significant metabolites (as above), and the relative concentration of the best individual discriminating metabolite (right). Data presented for; (**A**) *S.* Paratyphi A vs Controls, (AUC scores: 1.0, AUC CV scores: 0.999, AUC best metabolite: 0.884). (**B**) *S.* Typhi vs Controls (AUC scores: 1.0, AUC CV scores: 0.996, AUC best metabolite: 0.925). (**C**) *S.* Paratyphi A vs *S.* Typhi (AUC scores: 0.951, AUC CV scores: 0.898, AUC best metabolite: 0.693. Error bars represent mean score values with 95% confidence intervals.

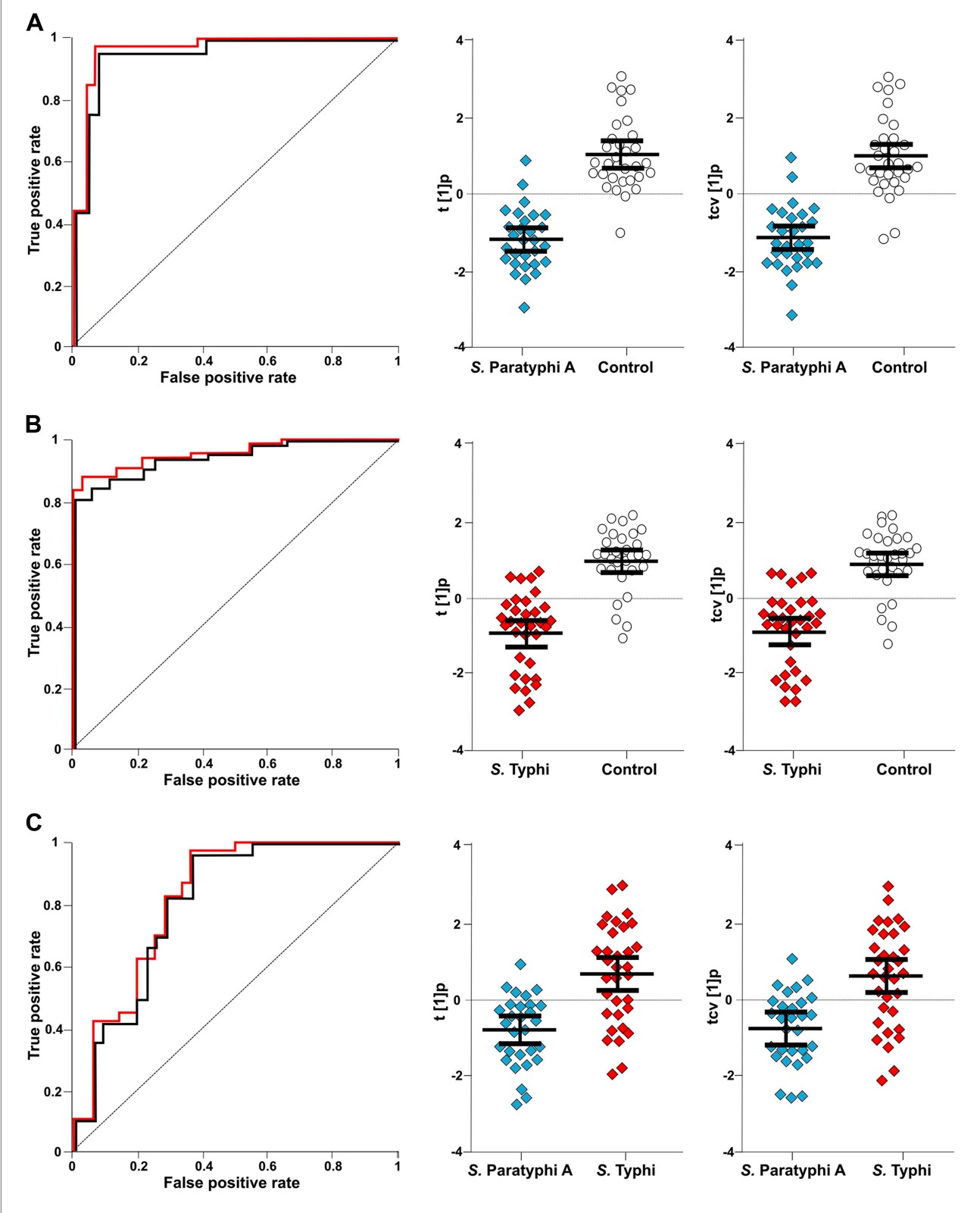

**Figure 6**. The discriminatory power of six metabolites to distinguish between plasma samples from controls, *S*. Typhi cases, and *S*. Paratyphi A cases. The panels on the left show the ROC-curves based on scores (red lines) and cross-validated scores (black lines) from OPLS-DA models using the six most statistically significant (*S*. Typhi against controls and/or *S*. Paratyphi A against controls) metabolites separating enteric fever samples from control

*Figure 6. Continued on next page*

*Figure 6. Continued*

samples and separating *S.* Typhi samples from *S.* Paratyphi A samples. The scatterplots show pairwise class differences based on scores (t[1]p) (left), cross-validated scores (tcv[1]p) (right) from OPLS-DA models using the 6 most statistically significant metabolites (as above). Data presented for; (**A**) *S.* Paratyphi A vs Controls, (AUC scores: 0.964, AUC CV scores: 0.948). (**B**) *S.* Typhi vs Controls (AUC scores: 0.934, AUC CV scores: 0.923) and (**C**) *S.* Paratyphi A vs *S.* Typhi (AUC scores: 0.801, AUC CV scores: 0.796). Error bars represent mean score values with 95% confidence intervals.

participants provided written informed consent for the collection and storage of all samples and subsequent data analysis, written informed consent was given for all those under 18 years of age by a parent or guardian (*Arjyal et al., 2011*).

## Study site and population

This study was conducted at Patan Hospital in Kathmandu, Nepal. Patan Hospital is a 318-bed government hospital providing emergency and elective outpatient and inpatient services located in Lalitpur Sub-metropolitan City (LSMC) within the Kathmandu Valley. Enteric fever is common at the outpatient clinic at Patan Hospital (*Karkey et al., 2010*; *Baker et al., 2011*), which has approximately 200,000 outpatient visits annually. The population of LSMC is generally poor, with most living in overcrowded conditions and obtaining their water from stone spouts or sunken wells.

The samples used for this study were collected from patients enrolled in a randomized controlled trial comparing gatifloxacin against ofloxacin for the treatment of uncomplicated enteric fever (ISRCTN 53258327) (*Arjyal et al., 2011*). The enrolment criteria were as previously described (*Pandit et al., 2007*). Briefly, patients who presented to the outpatient or emergency department of Patan Hospital, Lalitpur, Nepal from May 2009, to August 2011 with fever for more than 3 days who were clinically diagnosed to have enteric fever (undifferentiated fever with no clear focus of infection on preliminary physical exam and laboratory tests) whose residence was in a predesigned area of 20 km$^2$ in urban Lalitpur and who gave fully informed written consent were eligible for the study. Exclusion criteria were pregnancy or lactation, age under 2 years or weight less than 10 kg, shock, jaundice, gastrointestinal bleeding, or any other signs of severe typhoid fever, previous history of hypersensitivity to either of the trial drugs, or known previous treatment with chloramphenicol, a quinolone, a third generation cephalosporin, or a macrolide within 1 week of hospital admission.

## Microbiological culture and identification

Anti-coagulated blood samples were collected from all febrile patients upon arrival in the outpatient department. For those over the age of 12 years, 10 ml of blood sample was collected; 5 ml was collected from those aged 12 years or less. The blood samples were inoculated into tryptone soya broth and sodium polyethanol sulphonate up to 50 ml. The inoculated media was incubated at 37°C and examined daily for bacterial growth over seven days. On observation of turbidity, the media was subcultured onto MacConkey agar. Any bacterial growth presumptive of *S.* Typhi or Paratyphi was identified using serogroup specific antisera (02, 09, Vi) (Murex Biotech, Dartford, UK).

## Plasma samples

Two ml of peripheral blood was collected from all participants in sodium citrate tubes and were mixed well before being separated by centrifugation at 1000×*g* relative centrifugal force (RCF) for 15 min. The plasma and cells were separated before immediate storage at −80°C. Prior to metabolite analysis, 50 culture positive (25 *S.* Typhi and 25 *S.* Paratyphi A) plasma samples (with available patient metadata) were randomly selected from individual patients between the age of 12 and 22 years to in cooperate the median ages of both *S.* Typhi and *S.* Paratyphi A infections (*Karkey et al., 2010*). Additionally, 25 plasma samples from an age-stratified plasma bank gathered from patients attending Emergency Department of the Patan Hospital for reasons other than febrile illness throughout the same period and within the same 10-year age range as previously described were randomly selected for comparison (*Karkey et al., 2013*). The blood samples from these patients were collected, separated and stored as outlined above.

## Sample preparation for metabolomic analysis

The 75 plasma samples were divided into two batches that were maintained throughout the analysis process (in a random order but taking the sample parameters into consideration). The sample

containers were labeled with numbers to avoid awareness of sample group allocation during the sample preparation. All investigators were blinded to the source group of the plasma samples. The plasma samples were extracted and processed according to the plasma protocol for metabolomics at the Swedish Metabolomics Centre (SMC) (*Jiye et al., 2005*). Frozen 100 µl aliquots of plasma, in micro centrifuge tubes (Sarstedt Ref: 72.690), were thawed at room temperature and then kept on ice. Metabolite extraction was performed by addition of 900 µl methanol/water extraction mix (90:10 vol/vol) (including 11 isotopically labeled internal standards [7 ng/µl]) followed by rigorous agitation at 30 Hz for 2 min in a bead mill (MM 400; Retsch GmbH, Haan, Germany) and storage on ice for 120 min before centrifugation at 14,000 rpm for 10 min at 4°C (Centrifuge 5417R, Eppendorf, Hamburg, Germany). 200 microliters of each supernatant were transferred to gas chromatography (GC) vials and evaporated until dry in a speedvac (miVac; Quattro concentrator, Barnstead Genevac, Ipswich, UK). After evaporation the samples were stored in −80°C until derivatization. Prior to derivatization the extracted plasma samples were again dried briefly in a speedvac. Methoxyamination, by the addition of 30 µl methoxyamine in pyridine (15 µg/µl), 10 min of shaking and 60 min heating at 70°C, was carried out over 16 hr (at ambient temperature). Trimethylsilylation, with addition of 30 µl MSFTA (N-methyl-N-trimethylsilyl-trifluoroacetamide) + 1% TMCS (Trimethylchlorosilane), was performed for 1 hr (at ambient temperature). Finally, 30 µl heptane, including methyl stearate (15 ng/µl), was added as an injection standard.

## Metabolomic analysis by GCxGC/TOFMS

The two dimensional chromatography provides an output which can be seen as a metabolite landscape where each detected potential metabolite is defined by a three-dimensional peak in this landscape (retention time 1 × retention time 2 × peak height) (as shown in *Figure 1*). Extracted and derivatized plasma samples were analyzed, in a random order (within the analytical batches), on a Pegasus 4D (Leco Corp., St Joseph, MI, USA) equipped with an Agilent 6890 gas chromatograph (Agilent Technologies, Palo Alto, GA, USA), a secondary gas chromatograph oven, a quad-jet thermal modulator, and a time-of-flight mass spectrometer. Leco´s ChromaTOF software was used for setup and data acquisition. The column set used for the GCxGC separation was a polar BPX-50 (30 m × 0.25 mm × 0.25 µm; SGE, Ringwood, Australia) as first-dimension column and a non-polar VF-1MS (1.5 m × 0.15 mm × 0.15 µm; J&W Scientific Inc., Folsom, CA, USA) for the second-dimension column. Splitless injection of 1 µl sample aliquots was performed with an Agilent 7683B auto sampler at an injection temperature of 270°C (2 respectively 5 pre/post-wash cycles were used with hexane). The purge time was 60 s with a rate of 20 ml/min and helium was used as carrier gas with a flow rate of 1 ml/min. The temperature program for the primary oven started with an initial temperature of 60°C for 2 min, followed by a temperature increase of 4°C/min up to 300°C and where the temperature was held for 2 min. The secondary oven maintained the same temperature program but with an offset of +15°C compared to the primary oven. The modulation time was 5 s with a hot pulse time of 0.8 s and a 1.7 s cooling time between the stages. The MS transfer line had a temperature of 300°C and the ion source 250°C. 70 eV electron beams were used for the ionization and masses were recorded from 50 to 550 m/z at a rate of 100 spectra/sec with the detector voltage set at 1780 V. 15 randomly selected plasma samples were unblended and run in triplicate as analytical replicates (Control: N = 4, *S.* Paratyphi A: N = 5, *S.* Typhi: N = 6). In addition to the plasma samples, several samples of methyl stearate in heptane (5 ng/µl) were run to check the sensitivity of the instrument and three n-alkane series (C8-C40) were also run to allow calculation of retention indexes, RI. The analysis time was approximately 70 min/sample.

## Chemicals

All chemicals and compounds were of analytical grade unless stated otherwise. The isotopically labeled internal standards (IS) [$^2$H$_7$]-cholesterol, [$^{13}$C$_4$]-disodium α-ketoglutarate, [$^{13}$C$_5$,$^{15}$N]-glutamic acid, [1,2,3-$^{13}$C$_3$]-myristic acid, [$^{13}$C$_5$]-proline, and [$^2$H$_4$]-succinic acid were purchased from Cambridge Isotope Laboratories (Andover, MA, USA); [$^{13}$C$_4$]-palmitic acid (Hexadecanoic acid), [$^2$H$_4$]-butanediamine·2HCl (Putrescine), and [$^{13}$C$_{12}$]-sucrose from Campro (Veenendaal; [$^{13}$C$_6$]-glucose from Aldrich [Steinheim, Germany], The Netherlands); and [$^2$H$_6$]-salicylic acid from Icon (Summit, NJ, USA). Silylation grade pyridine and *N*-Methyl-*N*-trimethylsilyltrifluoroacetamide (MSTFA) with 1% trimethylchlorosilane (TMCS) were purchased from Pierce Chemical Co (Rockford, IL, USA). The stock solutions for reference compounds and IS were all prepared in 0.5 µg/µl concentrations in either Milli-Q water or methanol.

## Data processing and metabolite identification

Leco's ChromaTOF software was used for baseline correction, peak detection, mass spectrum deconvolution, mass spectra library search for identification and calculation of peak height/area. A signal-to-noise ratio of 10 was used for peak picking. The library search was performed against publicly available mass spectral libraries from US National Institute of Science and Technology (NIST) and from the Max Planck Institute in Golm (http://csbdb.mpimp-golm.mpg.de/csbdb/gmd/gmd.html) together with in-house libraries established at SMC. Peak information for each of the samples was exported as individual csv-files (comma-separated values). All csv-files were imported into the data processing software Guineu (1.0.3 VTT; Espoo, Finland) (*Castillo et al., 2011*) for alignment, normalization (with internal standards), filtering and functional group identification. After processing in Guineu all peaks were manually investigated by using the average spectra information, obtained from Guineu, in NIST MS Search 2.0 to search against the same libraries as previously used. This manual comparison was performed to additionally confirm the putative annotations of the metabolites and detect possible split peaks, which, if having comparable mass spectra and retention indices, were summed and compared to the individual peaks in the following multivariate statistical analysis to make decision about inclusion. During manual investigation, peaks were excluded from further analysis if detected in less than 50 samples, being an internal standard or silyl artifact, having few mass fragments in spectra, having mass spectra similar to another peak with a better identity match or being part of a sum. Metabolites found in less than 50 samples but still showing interesting profiles as diagnostic markers were interpreted separately.

## Pattern recognition

Pattern recognition is based on the concept of multivariate projection methods. In metabolomics pattern recognition is used to reduce the high dimensionality of acquired analytical data for facilitated interpretation of biochemical profile alterations and detection of patterns among characterized samples based on similarities in these biochemical profiles (*Holmes and Antti, 2002*; *Madsen et al., 2010*). Among multivariate projection methods principal components analysis (PCA) (*Wold et al., 1987*) and partial least squares (PLS) with its extension orthogonal-PLS (OPLS) are the most commonly applied for pattern recognition in metabolomics studies. Here PCA was used initially to obtain an overview of main variations in the acquired GCxGC/TOFMS data and to detect and remove outliers. To reduce confounding from analytical drift over the time of analysis PLS was used to fit a model with run order as the response, metabolites showing a strong correlation with run order (i.e., Pearson product moment correlation coefficient > $|0.5|$) were excluded from further modeling. OPLS with class information (for example if a plasma sample has been sampled from a non-infected control or an infected patient) as the response was then performed to detect metabolite patterns that best discriminate between the pre-defined sample classes. This type of pattern recognition modeling is referred to as discriminant analysis (DA), thus the method used is OPLS-DA (*Bylesjö et al., 2006*). OPLS-DA models were calculated in turn for (i) separation between the three sample classes (control, *S*. Typhi infected, and *S*. Paratyphi A infected), and (ii) for pairwise comparisons (control vs *S*. Typhi, control vs *S*. Paratyphi A, and *S*. Typhi vs *S*. Paratyphi A). For each model a $Q^2$ value was calculated to reflect the predictive power of the OPLS model. In the case of a DA model the $Q^2$ value, which can vary on a continuous scale between 0 and 1, will indicate if the classification (or metabolite pattern) is robust. A $Q^2$ of 1 refers to a perfect classification, while a $Q^2$ of 0 or below refers to a poor or random classification. In addition, a p-value was calculated for each OPLS-DA model using ANOVA (*Eriksson and Trygg, 2008*). To define which metabolites that contribute significantly to the detected metabolite patterns the OPLS-DA variable weights (covariance loadings; w*) and univariate p-values (two-tailed Student's *t* test) were used in combination. A metabolite was considered significant if it had a univariate p-value≤0.05 and was important for class separation in the OPLS-DA model, according to the variable weight or covariance loading (w*) (here the significance limit was w* >$|0.03|$ for the models separating non-infected controls and enteric fever samples and w* >$|0.07|$ for models between *S*. Typhi and *S*. Paratyphi A).

All pattern recognition analysis was performed in SIMCA (version SIMCA-P+ 13.0; Umetrics, Umeå, Sweden). Model plots were created using SIMCA or GraphPad Prism (5.04; GraphPad Software Inc., La Jolla, CA, USA) in combination with Adobe Illustrator CS5 (15.0.0; Adobe Systems Inc., San Jose, CA, USA).

Receiver operating curves (ROC) were constructed and compared for individual metabolites as well as for OPLS-DA model scores (metabolite profiles) to additionally investigate the usefulness of the

obtained results. The area under the curve (AUC) can be used as an output of the ROC analysis, which can range from 0.5 to 1.0. The higher AUC value a biomarker obtains the higher is the diagnostic potential. Here the web-based online tool ROCCET (http://www.roccet.ca/ROCCET/) was used to perform univariate ROC analyses. For the individual metabolites the relative concentrations for all samples were used as input, while for the models (metabolite profiles) model scores (t) and cross-validated scores (tcv) (*Stone, 1974*) were used after recalculation by subtracting the lowest score value from all other score values to avoid negative values.

## Acknowledgements

The authors wish to thank all the unit staff at the Patan Hospital in Kathmandu for assisting in sample, data collection and patient care. Peter Haglund and Konstantinos Kouremenos are acknowledged for their valuable input regarding the GCxGC/TOFMS analysis. Stephen Baker is a Sir Henry Dale Fellow, jointly funded by the Wellcome Trust and the Royal Society (100087/Z/12/Z). Henrik Antti is funded by the Swedish Research Council (VR-NT 2010-4284).

## Additional information

### Funding

| Funder | Grant reference number | Author |
|---|---|---|
| Wellcome Trust | 100087/Z/12/Z | Sabina Dongol, Abhilasha Karkey, Tuyen Ha Thanh, Stephen Baker |
| Royal Society | 100087/Z/12/Z | Sabina Dongol, Abhilasha Karkey, Tuyen Ha Thanh, Stephen Baker |
| Swedish Research Council | VR-NT 2010-4284 | Elin Näsström, Henrik Antti |

The funders had no role in study design, data collection and interpretation, or the decision to submit the work for publication.

### Author contributions

EN, Acquisition of data, Analysis and interpretation of data, Drafting or revising the article; NTVT, SD, AK, PVV, THT, AA, CD, BB, Acquisition of data, Contributed unpublished essential data or reagents; AJ, Conception and design, Drafting or revising the article; GT, Contributed unpublished essential data or reagents; SB, HA, Conception and design, Acquisition of data, Analysis and interpretation of data, Drafting or revising the article

### Ethics

Human subjects: The institutional ethical review boards of Patan Hospital and The Nepal Health Research Council and the Oxford Tropical Research Ethics Committee in the United Kingdom approved this study. All adult participants provided written informed consent for the collection and storage of all samples and subsequent data analysis, written informed consent was given for all those under 18 years of age by a parent or guardian (*Arjyal et al., 2011*).

## Additional files

### Supplementary file

• Supplementary file 1. Statistically significant metabolites in pairwise comparisons.

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
