## [Decision Letter]

Thank you for sending your work entitled “*Salmonella* Typhi and *Salmonella* Paratyphi A elaborate distinct systemic metabolite signatures during enteric fever” for consideration at *eLife*. Your article has been favorably evaluated by a Senior editor and 3 reviewers, one of whom, Quarraisha Abdool Karim, is a member of our Board of Reviewing Editors, and we are very pleased to inform you that your article has been accepted for publication.

The authors utilize Metabolomics to identify biomarkers to differentiate causes of enteric fever by *Salmonella* Typhi and Paratyphi, a major bottleneck to the development of new diagnostics. Enteric fever is a substantial public health problem in resource-constrained settings. The current diagnostic tool is blood culture but it has low specificity. Accurate diagnosis of enteric fever is important particularly in settings where febrile disease from multiple aetiologies is common and notably in the context of increased microbial resistance to inform appropriate treatment.

The application of metabolomics is relatively new in infectious diseases research. In this study specimens were carefully selected 75 patients with 50 from blood culture confirmed enteric fever patients (25 with *S.* Typhi and 25 with *S.* Paratyphi A) and 25 age range matched afebrile controls. Supervised pattern recognition was used to identify significant and reproducible metabolite profiles separating out Typhi, Paratyphi and controls following mass spectrometric analysis. A “fingerprint” of 6 metabolites that help differentiate the aetiological agent of enteric fever is defined. The authors underscore that while mass spectrometry cannot be used routinely the “signature” 6 metabolites could be used to inform the development of point of care diagnostics. They further highlight the potential for use in diagnosing other systemic bacterial infections using this methodology. These findings are a substantial technological advance to solve the problem of diagnosing enteric fever. I have no substantive or significant concerns with this manuscript and recommend acceptance for publication.

The manuscript is well written. However, there are a variety of abbreviations used that are not described before and it would help the reader understand the text if they were introduced. GCxGC/TOFMS, OPLS-DA, PCA, ROC, AUC, HCV, LSMC, and several more in the Methods section. In the Discussion kindly elaborate more about future applicability of this approach in the field. The reviewers also think that it would strengthen your story if you mention why patients with other febrile conditions were excluded.

[Editors’ note: given the minor nature of the comments, there is not an accompanying Author response]